# Development and Validation of an Analytical HPLC Method to Assess Chemical and Radiochemical Purity of [^68^Ga]Ga-NODAGA-Exendin-4 Produced by a Fully Automated Method

**DOI:** 10.3390/molecules27020543

**Published:** 2022-01-15

**Authors:** Silvia Migliari, Antonino Sammartano, Marti Boss, Martin Gotthardt, Maura Scarlattei, Giorgio Baldari, Claudia Silva, Riccardo C. Bonadonna, Livia Ruffini

**Affiliations:** 1Nuclear Medicine Division, Azienda Ospedaliero-Universitaria of Parma, 43126 Parma, Italy; ansammartano@ao.pr.it (A.S.); mscarlattei@ao.pr.it (M.S.); gbaldari@ao.pr.it (G.B.); lruffini@ao.pr.it (L.R.); 2Department of Medical Imaging, Radboudumc, 6500 HB Nijmegen, The Netherlands; Marti.Boss@radboudumc.nl (M.B.); martin.gotthardt@radboudumc.nl (M.G.); 3Food and Drug Sciences Department, University of Parma, Parco Area delle Scienze 27/A, 43126 Parma, Italy; claudia.silva@unipr.it; 4Department of Medicine and Surgery, University of Parma, 43126 Parma, Italy; riccardo.bonadonna@unipr.it; 5Division of Endocrinology and Metabolic Diseases, Azienda Ospedaliero-Universitaria of Parma, 43126 Parma, Italy

**Keywords:** [^68^Ga]Ga-NODAGA-exendin-4, GLP-1R, insulinoma, type 2 diabetes, automation

## Abstract

**Background:** Glucagon-like peptide 1 receptor (GLP-1R) is preferentially expressed in pancreatic islets, especially in β-cells, and highly expressed in human insulinomas and gastrinomas. In recent years several GLP-1R–avid radioligands have been developed to image insulin-secreting tumors or to provide a tentative quantitative in vivo biomarker of pancreatic β-cell mass. Exendin-4, a 39-amino acid peptide with high binding affinity to GLP-1R, has been labeled with Ga-68 for imaging with positron emission tomography (PET). Preparation conditions may influence the quality and in vivo behavior of tracers. Starting from a published synthesis and quality controls (QCs) procedure, we have developed and validated a new rapid and simple UV-Radio-HPLC method to test the chemical and radiochemical purity of [^68^Ga]Ga-NODAGA-exendin-4, to be used in the clinical routine. **Methods:** Ga-68 was obtained from a ^68^Ge/^68^Ga Generator (GalliaPharma^®^) and purified using a cationic-exchange cartridge on an automated synthesis module (Scintomics GRP^®^). NODAGA-exendin-4 contained in the reactor (10 µg) was reconstituted with HEPES and ascorbic acid. The reaction mixture was incubated at 100 °C. The product was purified through HLB cartridge, diluted, and sterilized. To validate the proposed UV-Radio-HPLC method, a stepwise approach was used, as defined in the guidance document released by the International Conference on Harmonization of Technical Requirements of Pharmaceuticals for Human Use (ICH), adopted by the European Medicines Agency (CMP/ICH/381/95 2014). The assessed parameters are specificity, linearity, precision (repeatability), accuracy, and limit of quantification. Therefore, a range of concentrations of Ga-NODAGA-exendin-4, NODAGA-exendin-4 (5, 4, 3.125, 1.25, 1, and 0.75 μg/mL) and [^68^Ga]Ga-NODAGA-exendin-4 were analyzed. To validate the entire production process, three consecutive batches of [^68^Ga]Ga-NODAGA-exendin-4 were tested. **Results:** Excellent linearity was found between 5–0.75 μg/mL for both the analytes (NODAGA-exendin-4 and ^68^Ga-NODAGA-exendin-4), with a correlation coefficient (R^2^) for calibration curves equal to 0.999, average coefficients of variation (CV%) < 2% (0.45% and 0.39%) and average per cent deviation value of bias from 100%, of 0.06% and 0.04%, respectively. The calibration curve for the determination of [^68^Ga]Ga-NODAGA-exendin-4 was linear with a R^2^ of 0.993 and CV% < 2% (1.97%), in accordance to acceptance criteria. The intra-day and inter-day precision of our method was statistically confirmed using 10 μg of peptide. The mean radiochemical yield was 45 ± 2.4% in all the three validation batches of [^68^Ga]Ga-NODAGA-exendin-4. The radiochemical purity of [^68^Ga]Ga-NODAGA-exendin-4 was >95% (97.05%, 95.75% and 96.15%) in all the three batches. **Conclusions:** The developed UV-Radio-HPLC method to assess the radiochemical and chemical purity of [^68^Ga]Ga-NODAGA-exendin-4 is rapid, accurate and reproducible like its fully automated production. It allows the routine use of this PET tracer as a diagnostic tool for PET imaging of GLP-1R expression in vivo, ensuring patient safety.

## 1. Introduction

Radiolabelled glucagon-like peptide 1 analogues, including exendin-3 and exendin-4, are endowed with strong potential for the clinical use in detection of insulinomas [1,2,3,4,5,6,7,8,9,10,11,12,13] as well as for non-invasive in vivo detection of pancreatic and transplanted islets of Langerhans in people with diabetes mellitus [14].

Initial studies with radiolabelled exendin derivatives were performed using In-111 and Tc-99m radionuclides for imaging with single photon emission computed tomography (SPECT) [2,3,4,9,10,15]. However, positron emission tomography (PET) has inherent advantages over SPECT in terms of higher sensitivity and spatial resolution as well as accurate quantification [16], which is crucial, especially considering the small size of pancreatic targets (insulinomas and beta cell mass). Among PET radionuclides Ga-68 is most promising in terms of its ready availability from a generator system, straightforward labelling chemistry, and favorable decay characteristics.

The production and accessibility of a radiopharmaceutical is one of the critical factors in PET imaging for both clinical trials and routine clinical examinations. [^68^Ga]Ga-NODAGA-exendin-4 was initially prepared manually [1], but for a routine production manual preparation would not be acceptable due to possible production variability and high radiation dose to the operator. Thus, automation of the synthesis process is desired in order to reduce radiation exposure, to improve tracer manufacturing robustness and to be compliant with GMP requirements for batch documentation. Once the synthesis is adequately optimized, the process and the final product need be validated (a regulatory requirement) ensuring clinical suitability and the adherence to well-defined standard operation procedures (SOPs) [17,18,19]. Indeed, validation and qualification activities are integral part of the radiopharmacy routine [20], and guidance documents are published and continuously updated by ICH [21,22], EDQM [23], EANM, and IAEA/WHO [24,25,26]. Radiochemical purity (RCP) is one of the most important quality criteria to release the final product for the clinical use, as described in the European Pharmacopeia [17,18,19]. To this aim, a validated separation method has to be available, enabling optimal separation between different (radio)chemical forms (radioactive impurities) other than the original intact radiopharmaceutical [27,28,29].

Many radiopharmaceuticals, such as [^68^Ga]Ga-NODAGA-exendin-4, are produced as small-scale preparations [30] and general guidelines on quality controls (QCs) must be applied for clinical use. 

Here, we would like to introduce the fully automated synthesis conditions of [^68^Ga]Ga-NODAGA-exendin-4 moving to optimization of published research achievements [31,32] and our previous encouraging results on a semi-manual synthesis method [33,34,35]. The proposed fully automated synthesis method concerns the use of a more concentrated HEPES buffer solution together with ascorbic acid, higher labelling temperature and longer incubation time, as well as a more efficient purification step. 

Then we validated/qualified a practical QC system, including high-pressure liquid chromatography (HPLC), to assess chemical and radiochemical purity of the final product in accordance with Ph. Eur. (9.0/0125, monographs 2482 and 2464) and the acceptance criteria derived from published data [36,37,38].

## 2. Results

### 2.1. Validation of the UV-Radio-HPLC Method

Under the above described chromatographic conditions, NODAGA-exendin-4, Ga-NODAGA-exendin-4 and all the other peaks were well resolved.

In Figure 1 typical chromatograms of blank eluent are illustrated in comparison to spiked samples. The average retention time (tR) of NODAGA-exendin-4 and Ga-NODAGA-exendin-4 was 6.852 ± 0.7 min and 6.755 ± 0.7 min, respectively.

The calibration curve for the determination of NODAGA-exendin-4 and Ga-NODAGA-exendin-4 was linear over the range 5–0.75 µg/mL. The linearity of this method was statistically confirmed for each calibration curve (Figure 2). 

The R^2^ for calibration curves was equal to 0.999, in accordance with acceptance criteria (Table 1). 

For each point of calibration standard, the concentrations of NODAGA-exendin-4 and Ga-NODAGA-exendin-4 were recalculated from the equation of linear regression curve. CV% was less than 2%, while the bias % values did not deviate more than 5% at all concentrations of both analytes. All data are provided in Appendix A. The LOQ for NODAGA-exendin-4 and Ga-NODAGA-exendin-4 was 0.75 µg/mL.

Under the same chromatographic conditions, [^68^Ga]Ga-NODAGA-exendin-4 (Figure 3) and all the other peaks were well resolved for all three validation batches. 

The calibration curve for the determination of [^68^Ga]Ga-NODAGA-exendin-4 was linear (Figure 4), the R^2^ for calibration curves was equal to 0.993 and the average CV% was <2% in accordance to acceptance criteria (Table 2).

The average values of the sample concentrations, as recalculated on the basis of the calibration line, the coefficient of variation (%) and the accuracy, expressed as bias%, are found in Appendix A.

### 2.2. Validation of the Process for Producing and Controlling [^68^Ga]Ga-NODAGA-Exendin-4

The radiochemical yield (RCY%), the molar activity (A_m_) and the RCP% of the final product were evaluated after the purification of [^68^Ga]Ga-NODAGA-exendin-4 on HLB cartridge, at the end of the synthesis, for all the three consecutive batches, as shown in Appendix A.

The UV-Radio-HPLC showed free Gallium-68, detected at retention time (tR) = 1.450 min, Ga-68 bound to NODAGA-exendin-4 at a mean tR of 6.955 ± 0.7 min and other irrelevant radioactive impurities at 5.647 and 11.517 min (Figure 3). 

The Radio-TLC detected no Ga-68 colloids (Rf < 0.2) and showed only the presence of [^68^Ga]Ga-NODAGA-exendin-4 (Rf > 0.4) (Figure 5). 

The RCP% tested by UV-Radio-HPLC and Radio-TLC was 97.05%, 95.75%, and 96.15% for the three validation batches, respectively (mean value 96.31%). 

The pH of the radiopharmaceutical was 7 in all the validation runs. The endotoxin concentration was <17.5 IU/mL (respectively, 4.85 EU/mL, 4.80 EU/mL, and 4.95 EU/mL, and 97.00%, 96.10%, and 97.36% for the spike recovery). All products resulted sterile.

The Ge-68 breakthrough determined in the three samples was constantly well below the level recommended by Ph. Eur. (0.001% of the total radioactivity) and, respectively, 3.5 × 10^−7^%, 3.3 × 10^−7^%, and 3.5 × 10^−7^%. As to the radionuclide identity of Ga-68, the spectrum obtained by gamma spectroscopy peaked at energies equal to 0.511 MeV and 1.077 MeV corresponding to Ga-68, data confirmed also by the calculation of the half-life of the radionuclide (mean value 64.2 min). We also documented the residual quantity of HEPES in the final preparation by Radio-TLC. The HEPES spot was less intense than that corresponding to the reference solution according to Ph. Eur. (200 μg/V) in all the validation batches. Residual ethanol (5.23%, 6.55% and 5.68%) was within allowed limits.

Quality control results calculated in the three validation batches and acceptance criteria are reported in Table 3.

The stability of [^68^Ga]Ga-NODAGA-exendin-4 in PBS solution at room temperature was tested up to 4 h via Radio-TLC and UV-Radio-HPLC. [^68^Ga]Ga-NODAGA-exendin-4 is stable as confirmed by a RCP% values > 95% over the entire period (mean RCP% for the three synthesis runs: 96.76% at T0, 95.45% at T2 h, and 95.38% at T4 h) (Figure 6).

## 3. Discussion

In this article we present the results and validation of a fully automated synthesis method of [^68^Ga]Ga-NODAGA-exendin-4.

The increasingly call to optimize and standardize the production process to comply clinical needs for new tracers, prompted us to improve our previously developed semi-manual synthesis of [^68^Ga]Ga-NODAGA-exendin-4 [33,34,35]. Moreover, we have validated the specific QC system for the new radiotracer.

Considering our optimal results, the amount of starting peptide to produce [^68^Ga]Ga-NODAGA-exendin-4 was 10 µg (0.002 µmol).

The new automated synthesis method is different from the semi-manual one in many aspects. In the reaction mixture we used a more concentrated HEPES buffer (2.5 M) together with the radical scavenger ascorbic acid (100 mg/mL) to buffer better the acidity of the eluate and to reduce radiolysis of the peptide during the complexation with Ga-68. Additionally, we have incubated the reaction solution at higher temperature and longer time (100 °C, 15 min) to promote reaction pharmacokinetic.

Another important difference was the purification of [^68^Ga]Ga-NODAGA-exendin-4 on HLB cartridge, starting from an optimized preconditioning step with 1 mL Ethanol, 10 mL water, and air. Then, after the cooling of the radiopharmaceutical and before its loading on the column, a mixture of ethylenediaminetetraacetic acid (EDTA)/polysorbate 80 was added on the mixture solution in order to chelate any unbound Ga-68 and to prevent sticking of the [^68^Ga]Ga-NODAGA-exendin-4.

In Appendix A results of our optimized and automated synthesis were reported, showing the values of radiochemical yield, 45% (23.53%) and radiochemical purity 96.32% (91.69%). Moreover, the molecular activity A_m_ 33.30 GBq/µmol ± 0.3 (100 GBq/µmol) was maintained sufficiently high, enabling the injection of low amount of peptide (10 μg), thereby mitigating the pharmacologic effects of the compound [1,33,39,40,41,42]. Finally, the use of an automated technology such as the synthesis module, located inside a grade A hot cell, is an advantage in terms of radioprotection, standardization, and harmonization. Indeed, the hot cell allows the reduction of radiation exposure of the operator and the automation facilitates the adoption of our synthesis method by different centers.

To evaluate the chemical and radiochemical purity of [^68^Ga]Ga-NODAGA-exendin-4, we used the QC system already developed in our previous work [33,34,35].

Results of synthesis validation, summarized in Table 3, show that all the tested quality parameters were in accordance with the European Pharmacopoeia. Moreover RCP% remains >95% over a 4-h period (Figure 6) allowing the use of the radiotracer for image acquisition.

The higher values of RCP%, obtained with the fully automated method compared to our previous one, were confirmed with both radio-TLC and Radio-UV-HPLC CQs. The developed Radio-UV-HPLC led us to verify exactly the presence of free Ga-68 (1.450 min) and the radiopharmaceutical product (6.955 min), but also to detect other radioactive impurities at 5.647 and 11.517 min (Figure 3). We have integrated both these peaks and they did not affect the RCP% of the product that resulted more than 95% in all the three validation batches (97.05%, 95.75%, and 96.15%). Therefore, being that the impurity peak areas were very low, we considered it unnecessary and not essential to isolate and characterize them. The higher RCP% of the radiopharmaceutical was also confirmed with Radio-TLC method (Figure 5) that is able to detect Ga-68 colloids at Rf < 0.2 and [^68^Ga]Ga-NODAGA-exendin-4 at Rf > 0.4 (Appendix A). 

Finally, to verify the linearity and the precision of our UV-Radio-HPLC method we validated it and the results (Appendix A) demonstrate its robustness and reproducibility.

## 4. Materials and Methods

### 4.1. Reagents, Radionuclides

All chemicals used for the radiolabelling reaction were commercially obtained as a single disposable kit (reagents and cassettes for synthesis of ^68^Ga-peptides using cationic purification ABX, Advanced Biochemical Compounds, Radeberg, Germany). 

The peptide NODAGA-exendin-4 and Ga-NODAGA-exendin-4 were purchased as lyophilized powder from piCHEM (Forschungs und Entwicklungs, Grambach, Austria).

Ga-68 (t_1/2_ = 68 min, β^+^ = 89%, and EC = 11%) was obtained from a pharmaceutical grade ^68^Ge/^68^Ga generator (1850 MBq, GalliaPharm^®^ Eckert and Ziegler, Berlin, Germany) by elution with 0.1 M HCl (Rotem GmbH, Leipzig, Germany). The amount of detected metal impurities as provided by the manufacturer was less than the defined limit in the European Pharmacopeia monograph [35,36]. 

The reagents trifluoroacetic acid, acetonitrile, ammonium acetate, and dimethylformamide (DMF) used for QCs, such as mobile phases for UV-Radio-HPLC and instant thin layer chromatography (Radio-TLC), were metal-free and purchased from Sigma Aldrich (Saint Louis, MO, USA).

Stock reference solutions (5 μg/mL) and appropriate dilutions of NODAGA-exendin-4 and Ga-NODAGA-exendin-4 were prepared in ultrapure water (Sigma Aldrich) and stored at −20 °C. All chemicals used for both synthesis and QCs were of pure and analytical grade.

The aseptic production was conducted in a GMP grade A hot cell (NMC Ga-68, Tema Sinergie, Ravenna, Italy). Both ^68^Ge/^68^Ga generator and automated synthesis module (Scinotmics GRP^®^ module, Fürstenfeldbruck, Germany) were placed in the hot cell.

### 4.2. Automated Synthesis of [^68^Ga]Ga-NODAGA-Exendin-4

^68^Ga-NODAGA-exendin-4 was synthesized using a fully automated platform for labelling synthesis with disposable cassette system (SC-102, ABX). 

Reaction parameters as reaction time, temperature, and radioactivity were monitored in real time. The process included the pre-concentration of the generator elute through a strong cation exchange (SCX) cartridge, pre-conditioned with 10 mL 5 M NaCl in 0.1 M HCl solution, and then the same solution was used for the recovery of Ga-68 (III) from SCX cartridge into the reactor.

The reaction mixture contained 2.5 M 4-(2-hydroxyethyl)-1-piperazineethanesulfonic acid (HEPES), buffer, and the radical scavenger ascorbic acid (100 mg/mL) to reduce radiolysis. The precursor (0.05 μg/μL NODAGA-exendin-4) was incubated in the heating block for 15 min at 100 °C. 

After the reaction completion the crude product was cooled down and 2 mL of ethylenediaminetetraacetic acid (EDTA) 50 mM/polysorbate 80 0.15% were added to the reaction vial.

Then, [^68^Ga]Ga-NODAGA-exendin-4 was loaded on a hydrophilic–lipophilic balance (HLB) cartridge to be purified. The cartridge was pre-conditioned with 1 mL ethanol followed by 10 mL of water and 10 mL of air optimizing the already published method [31].

Finally, the radiopharmaceutical was eluted with 1 mL of ethanol, sterilized through a 0.2 μm filter (millex GV) into a sterile 25 mL capped glass vial and diluted with sterile 0.9% saline for the final formulation.

### 4.3. Quality Controls and Process Validation

After synthesis, the [^68^Ga]Ga-NODAGA-Exendin-4 solution was analyzed to assess the presence of radiochemical and chemical impurities, which may originate from different sources such as radionuclide impurities, radiolabeling procedure, incomplete preparative separation, or chemical changes of the molecule during storage. 

The QCs have to determine the following parameters: total product activity, ^68^Ga^3+^ identity via half-life time and gamma spectroscopy, chemical and radiochemical purity by UV-Radio-HPLC and Radio-TLC, pH, radionuclide purity for ^68^Ge-breakthrough, and sterility/endotoxin assay (sterility test and LAL test).

Moreover, the stability of [^68^Ga]Ga-NODAGA-exendin-4 at room temperature was monitored by Radio-TLC and UV-Radio-HPLC for 4 h.

To validate the entire process of radiopharmaceutical production and quality control, three batches of [^68^Ga]Ga-NODAGA-exendin-4 were produced in three different days under the same conditions set for typical routine preparations. Every batch was fully characterized from the analytical point of view, with the aim to verify that the product meets the acceptance criteria for all the established quality parameters. 

#### 4.3.1. Appearance

The visual inspection of in-house prepared radiopharmaceuticals is necessary before injection into the patient, as a measure of process performance and validation. The presence of particulate in the sample suggests possible failure during radiopharmaceutical synthesis, including purification, sterilizing filtration, and failed environmental control during the setting up of reagents [43].

#### 4.3.2. Instant Thin Layer Chromatography

Radio-TLC test was used to determine the percentage of [^68^Ga]Ga-NODAGA-exendin-4 and Ga-68 colloids in the final product. 

For the determination of 68Ga-colloids percentage, ammonium acetate 1 M pH 5.5 and DMF was used as mobile phase and TLC-SG paper strips (Varian TLC-SG plates) as stationary phase; Ga-68 colloids could be detected at Rf < 0.2, but the [^68^Ga]Ga-NODAGA-exendin-4 at Rf > 0.4. To verify the sensitivity and appropriateness of this TLC method we tested a solution of the radiopharmaceutical product together with the impurity. Appendix A shows exactly the detection of both two entities, results that were well separated and integrable.

TLC-SG paper strips, used a stationary phase, were counted with a scanner (Cyclone^®^ Plus Storage Phosphor system, Perkin Elmer, Milan, Italy) and the chromatograms were analysed with OptiQuantTM software.

#### 4.3.3. High-Pressure Liquid Chromatography

The UV-Radio-HPLC was additionally used to determine the percentage of [^68^Ga]Ga-NODAGA-exendin-4 in the final product after the purification with HLB cartridge, at the end of the synthesis. 

UV-Radio-HPLC was performed on a Dionex Ultimate 3000 HPLC system (Thermo Fisher Scientific, Waltham, MA, USA) equipped with an AcclaimTM 120 C18 column 3 µm 120 Å (3.0 mm × 150 mm) and a UV and a γ-detector (Berthold Technologies, Milan, Italy). The used solvents were A) 0.1% trifluoroacetic acid (TFA) in water and B) 0.1% TFA in acetonitrile. 

The flow rate of the mobile phase was set at 0.6 mL/min, with a total run of 15 min.

The following phase gradient was used in the UV-Radio-HPLC analysis: 0–2 min 5% B, 2–7 min from 5% B to 100% B, 7–8 min 100% B, 8–12 min from 100% B to 5% B, and 12–15 min 5% B.

The column temperature was kept at 30 °C. The samples were also monitored with UV detector at 220 nm in order to detect chemical impurities in the final product. Ga-68 ion and [^68^Ga]Ga-NODAGA-exendin-4 were measured by UV-Radio-HPLC γ-detector.

The software system Chromeleon 7 was used to assemble the information.

#### 4.3.4. Ge-68 Breakthrough

The Ge-68 breakthrough was measured by gamma spectroscopy of the final product, using a gamma spectrometer equipped with a high-purity germanium (HPGe detector ORTEC GEM 30P4-76). The *γ*-ray spectrometry tests included the identification of principal *γ*-photon (499–521 KeV peak) and Ge-68 content (decay of 499–521 KeV peak ≥ 48 h) using a large volume counter linked to a multichannel analyser system (HPGe detector ORTEC GEM 30P4-76).

The spectra acquisition was performed at least 48 h after the synthesis to allow the Ga-68 to decay to a level low enough to permit the detection of Ge-68. 

Duration of the acquisition was 180 min to obtain a high signal-to-noise ratio. The sample volume was at least 1 mL. Spectrum was analyzed using Genie 2000 software.

#### 4.3.5. Radionuclide Identification and Activity Measurements

Radionuclide purity was determined based on the half-lives, type, and energy of the emitted radiations. Half-life was measured with a dose calibrator (Capintec 25-R) at four consecutive intervals (5, 10, 15, and 20 min). The expected half-life of Ga-68 is 67.6 min and is calculated using the following equation: *T*_1/2_ = −*ln*2 (*dt*/*ln* (*A*_1_/*A*_0_))(1)
where: *dt*—time difference, *A*_1_—ending activity, and *A*_0_—starting activity. 

#### 4.3.6. pH Evaluation

The pH value of [^68^Ga]Ga-NODAGA-exendin-4 was measured using colorimetric pH strips (0–14). 

#### 4.3.7. Endotoxin and Sterility

Quantitative determination of bacterial endotoxins was performed by the chromogenic method, using Endosafe^®^ nexgenPTS™ (Charles River, MA, USA, Stati Uniti) apparatus. [^68^Ga]Ga-NODAGA-exendin-4 samples were previously diluted and then applied in duplicate inside cartridges in parallel with positive control testing. The radiopharmaceutical was considered apyrogenic when the level of endotoxins was less than 17.5 IU/mL in accordance with Ph. Eur. (9.0/0125).

The sterility of the [^68^Ga]Ga-NODAGA-exendin-4 solution was assessed by direct inoculation in a growth broth (Triptic Soy Broth, TSB) which was incubated at 20–25 °C, and verified daily over 14 days [44]. The sample was considered sterile when no microbiological growth was detected. 

#### 4.3.8. Residual Solvents and HEPES

Potentially present radiolysis products, such as ions and excited molecules, need be searched and removed, because they could cause undesired and serious side effects [45,46]. Radiolysis may be reduced by utilizing compounds insensitive to radiation or extenuating the process with additives (e.g., radical scavengers). In the clinical context, these radical scavengers should be suitable for human use, such as ascorbic acid or HEPES in the reaction mixture and ethanol in the pre- and post-processing steps. The positive influence of ethanol on radiolabeling yield and radiolysis restrain [47,48,49] prompted us to use it during the synthesis process in order to obtain a more reliable and repeatable automated method.

Ethanol is a class 3 solvent which may remain indeterminate and unmentioned up to 0.5%, but must be declared quantitatively for higher amounts in a pharmaceutical (Ph. Eur. 2019, 9.6. 9.7–9.8). 

In this study we determined the residual ethanol using gas chromatography (GC) and HEPES content by TLC-SG, both according to Ph. Eur. Monograph [37]. The reference solution of HEPES was prepared at a concentration of 200 µg/mL. Two separate spots of reference solution (5 µL) and test solution (a sample of the final product [^68^Ga]Ga-NODAGA-exendin-4) were applied on TLC silica gel F_254_ plate and developed on a path over 2/3 of the plate with use of water:acetonitrile (25:75 *v/v*) solution as a mobile phase. The plate was then exposed to iodine vapor. The spot corresponding to the test solution should not be more intense than the reference solution spot (less than 200 µg/V of HEPES in test solution).

### 4.4. Validation of UV-Radio-HPLC Method to Determine the Chemical Purity

Validation of the analytical method to determine the chemical purity of [^68^Ga]Ga-NODAGA-exendin-4 was carried out according to ICH Q2 (R1) and EDQM guidelines [20,21,23], which define type of analytical methods to be validated, set parameters, and acceptance criteria to be considered. Tests and acceptance criteria assessed in the validation process are listed in Table 1 [17].

#### 4.4.1. Specificity

Specificity is the ability of the analytical method to distinguish between the component of interest in its intended formulation and the other components in the final product. In order to demonstrate specificity, a series of solutions containing the critical components ([^68^Ga]Ga-NODAGA-exendin-4 and free Ga-68) at different concentrations were analyzed. 

#### 4.4.2. Linearity

Linearity is defined as the proportional response of a method as a function of the amount of analyte. Linearity is expressed by a linear regression calculated through the obtained results with the analyte at different concentrations within a pre-selected range. Determination of linearity was done on sets of standard solutions with different concentration of NODAGA-exendin-4 and Ga-NODAGA-exendin-4. Scalar solutions (5, 4, 3.125, 1.25, 1, and 0.75 µg/mL) were prepared by serial dilutions from a high concentration “mother” solution. Data were fitted by a least-squares regression method. The curve equation and the correlation coefficient (R^2^) are calculated through the equation *y* = *ax* + *b*, where *y* is peak area, *a* the slope, *x* the analyte concentration, and *b* the intercept.

#### 4.4.3. Precision and Accuracy

A measurement is precise if it obtains similar results with repeated measurements. Precision is usually expressed as the coefficient of variation (CV%) determined using the equation CV% = (*s*/*m*) × 100, where *m* is the average of the replicate measurements and *s* is the standard deviation.

We performed five replicate UV-Radio-HPLC injections of a standard sample solution at the nominal concentration of 1 nmol/mL. 

Accuracy is the degree of agreement between the result found by the analytical method and the true value. Accuracy is quantitatively expressed as bias (%), determined as the difference between the mean value obtained from replicate measurements and the reference value of a given sample ((average concentration observed/nominal concentration) × 100). Acceptance criteria is bias% > 95%. We prepared five replicate standard sample solutions at three concentration levels over the range of 50% to 150% of the nominal sample concentration of 1 nmol/mL. Assessment of precision included an intra-day (repeatability) and inter-day component estimated by analyzing six replicates at three different QC levels, i.e., 4.5, 2.0, and 0.85 μg/mL injected at consecutive times of the same day and at three alternative days. According to the ICH validation guideline CV% < 2% reflects high precision of the method.

Precision was assessed by the coefficient of variance (CV) of either 6 (intraday) or 18 (interday) independent replicates.

#### 4.4.4. Limit of Quantitation (LOQ)

The quantification limit is the lowest amount of analyte that gives a signal-to-noise ratio of at least 10:1 (the minimum concentration at which the sample can be reliably quantified). Experimental LOQ were determined by analyzing a series of diluted solutions of standard NODAGA-exendin and Ga-NODAGA-exendin-4 solutions, until the LOQ was reached. 

### 4.5. Validation of UV-Radio-HPLC Method to Determine the Radiochemical Purity

In Table 2 the validation parameters of the analytical method and their acceptance criteria are reported.

Some of the ICH validation parameters do not apply for radioactive compounds considering the radioactive nature and the short half-life of Ga-68.

#### 4.5.1. Linearity

The typical experimental approach based on the preparation of a series of solution with different concentrations cannot be applied to a radioactive compound. In this case one sample solution only, with a suitable radioactive concentration, was analyzed five times, at defined time intervals (15 min). Indeed, the radioactivity being the physical parameter of concern for radiochemical detectors, the radionuclide decay itself provides the necessary linear series of values. 

R^2^ was extrapolated from the calibration curve by analyzing the five different radioactive concentration of [^68^Ga]Ga-NODAGA-exendin-4. 

#### 4.5.2. Precision

The same considerations described for linearity are also valid for precision, as the radionuclide decay inevitably leads to a decrease over time of the radioactivity. However, repeatability may be evaluated analyzing a series of UV-Radio-HPLC runs obtained with repetitive injections of a single [^68^Ga]Ga-NODAGA-exendin-4 sample, and recalculating the obtained peak area values with the decay equation: *lnA*_0_ = *lnA* + *λt*, *where*
*λ* = 0.693/*t*_1/2_(2)
*A*_0_ = corrected peak area; *A* = measured peak area; *t* = time interval between the considered injection and the first one; and *t*_1/2_ = half-life (^68^Ga = 67.63 min).

The peak area values normalized for decay are compared and yield a consistent statistical analysis. Average standard deviation (SD) and CV% are then calculated. 

Repeatability has to be determined in three different days, to verify the instrument outcome during the time course.

## 5. Conclusions

The fully automated GMP/cGRPP compliant synthesis and the developed UV-Radio-HPLC method for QCs of [^68^Ga]-Ga-NODAGA-exendin-4 permits a reliable and highly reproducible production of the radiopharmaceutical improving standardization and harmonization of the entire process, allowing routine use of this PET tracer as diagnostic and research tool for imaging GLP-1R expression in vivo and enabling reproducible imaging results between different centers.

## Figures and Tables

**Figure 1 molecules-27-00543-f001:**
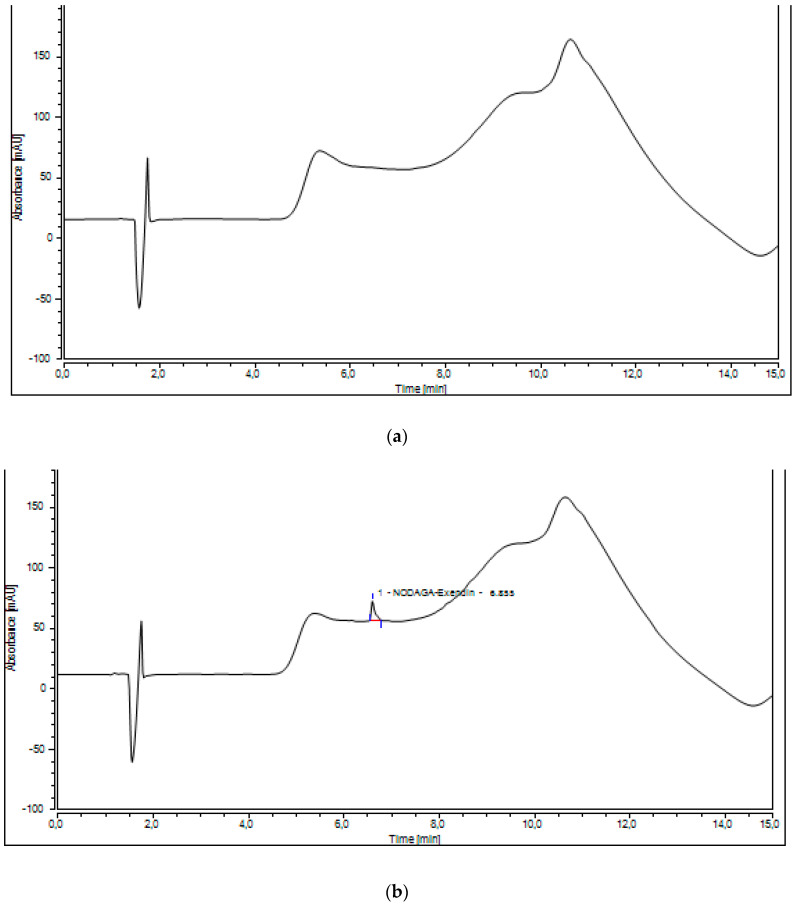
UV-Radio-HPLC traces of blank sample (**a**), NODAGA-exendin-4 (**b**), and Ga-NODAGA-exendin-4 (**c**). Pure and analytical grade eluents: 0.1% trifluoroacetic acid (TFA) in water (A)/0.1% TFA in acetonitrile (Bb), flow rate 0.6 mL min^−1^, column temperature 30 °C, wavelength of detection: 220 nm. The gradient elution profile: 0–7 min—5–100% B and 7–15 min—100–5% B.

**Figure 2 molecules-27-00543-f002:**
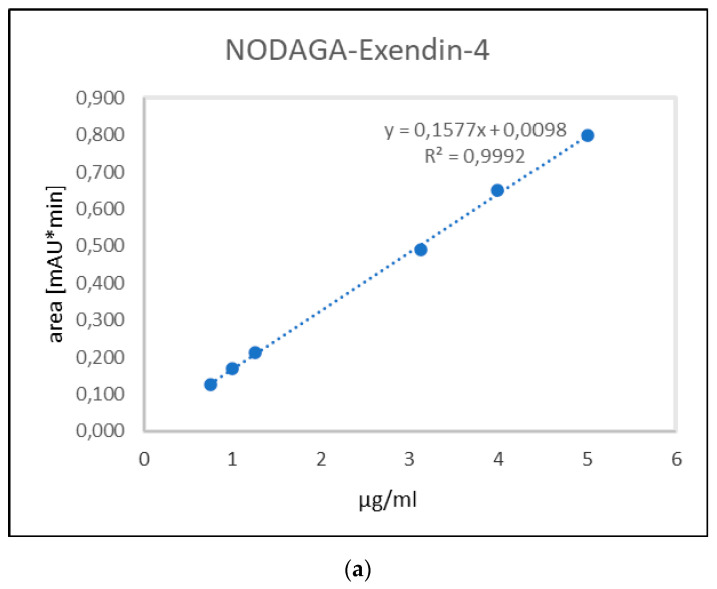
Calibration curve obtained with the average values of peak areas of six different concentrations (5, 4, 3.125, 1.25, 1, and 0.75 μg/mL) of NODAGA-exendin-4 (**a**) and Ga-NODAGA-exendin-4 (**b**), respectively. For each concentration of NODAGA-exendin-4 and Ga-NODAGA-exendin-4 the peak areas were measured and a mean value for each was calculated to obtain the calibration curve: 0.7986 ± 0.0020, 0.6495 ± 0.0022, 0.4893 ± 0.0029, 0.2117 ± 0.0010, 0.1704 ± 0.0014, 0.1251 ± 0.0002 mAU*min for NODAGA-exendin-4 and 0.7989 ± 0.0039, 0.6493 ± 0.0019, 0.4893 ± 0.0030, 0.2122 ± 0.0004, 0.1704 ± 0.0006, 0.1245 ± 0.0004 mAU*min for Ga-NODAGA-exendin-4.

**Figure 3 molecules-27-00543-f003:**
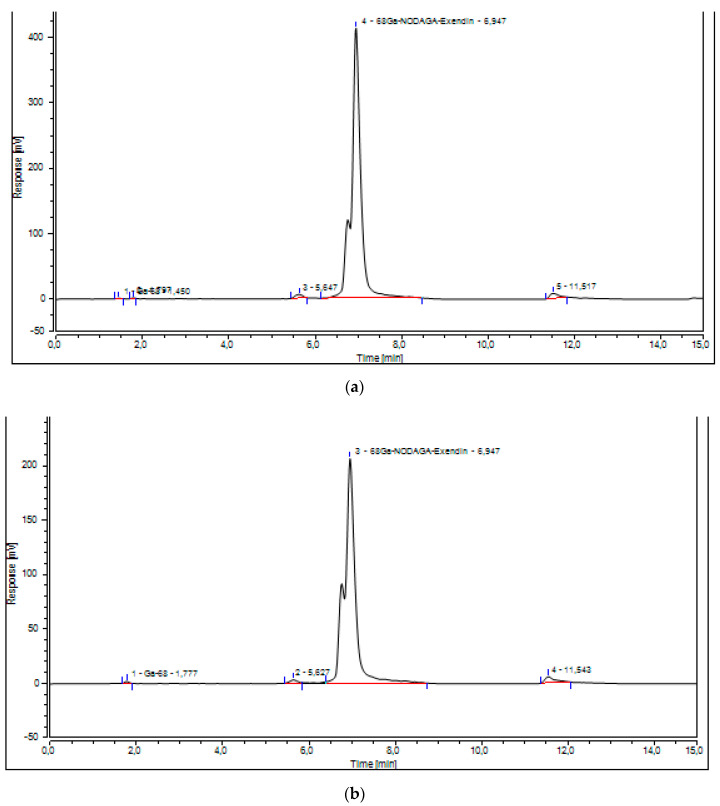
UV-Radio-HPLC chromatograms of [^68^Ga]Ga-NODAGA-exendin-4 for all three validation batches (**a**–**c**). Pure and analytical grade eluents: 0.1% trifluoroacetic acid (TFA) in water (A)/0.1% TFA in acetonitrile (B), flow rate 0.6 mL min−1, column temperature 30 °C, wavelength of detection: 220 nm. The gradient elution profile: 0–7 min—5–100% B and 7–15 min—100–5% B.

**Figure 4 molecules-27-00543-f004:**
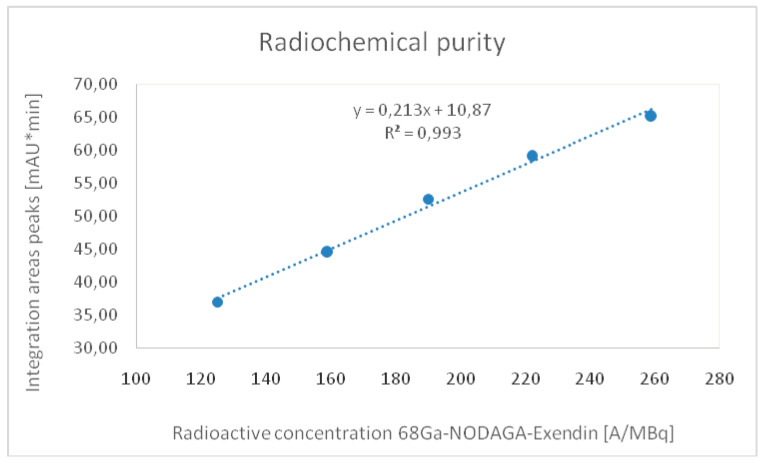
Calibration curve obtained with the average values of peak areas of [^68^Ga]Ga-NODAGA-exendin-4.

**Figure 5 molecules-27-00543-f005:**
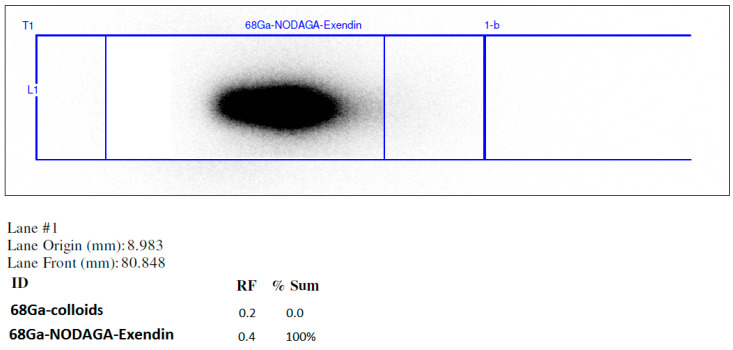
Radio-TLC of [^68^Ga]Ga-NODAGA-exendin-4 conducted with ammonium acetate 1M pH 5.5 and DMF.Ga-68 colloids could be detected at Rf < 0.2 and [^68^Ga]Ga-NODAGA-exendin-4 at Rf > 0.4 as showed in Appendix A.

**Figure 6 molecules-27-00543-f006:**
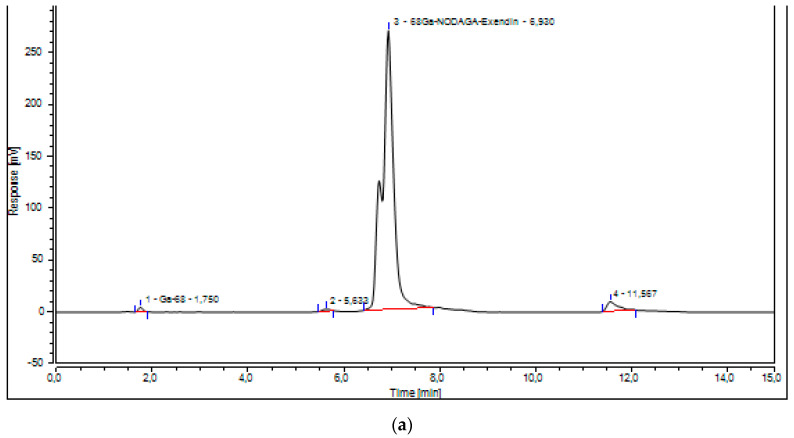
Stability of [^68^Ga]Ga-NODAGA-exendin-4 at T0 (**a**), T2 h (**b**), and T4 h (**c**).

**Table 1 molecules-27-00543-t001:** Tests and acceptance criteria in determining chemical purity using UV-Radio-HPLC.

Test	Acceptance Criteria
Specificity	≥2.5
Linearity	R^2^ ≥ 0.99
Repeatability	CV% < 2%
Quantification limit (LOQ)	CV% < 5%
Accuracy	bias% > 95%

**Table 2 molecules-27-00543-t002:** Tests and acceptance criteria in determining radiochemical purity using UV-Radio-HPLC.

Test	Acceptance Criteria
Specificity	Not applicable
Linearity	R^2^ ≥ 0.99
Repeatability	CV% < 2%
Quantification limit (LOQ)	Not applicable
Accuracy	Not applicable

**Table 3 molecules-27-00543-t003:** Summary of acceptance criteria and QC results calculated for all three validation batches of [^68^Ga]Ga-NODAGA-exendin-4, obtained from 10 µg of precursor.

Test	Acceptance Criteria	Batch 1	Batch 2	Batch 3
Radiochemical purity (UV-Radio-HPLC)	[^68^Ga]Ga-NODAGA-exendin-4 ≥ 90% ^68^Ga^3+^ and impurities < 10%	97.05% /	95.75% /	96.15% /
Radiochemical purity (Radio-TLC)	[^68^Ga]Ga-NODAGA-exendin-4 ≥ 90% ^68^Ga-colloids < 3%	100% /	100% /	100% /
pH	4–8.5	7	7	7
Radioactivity	50–500 MBq	331 MBq	333 MBq	336 MBq
Volume	2–16 mL	16 mL	16 mL	16 mL
Color	colorless	Colorless	Colorless	Colorless
Molar activity	1–150 GBq/µmol	33.10 GBq/µmol	33.30 GBq/µmol	33.60 GBq/µmol
Ge-68 breakthrough	<0.001%	0.00000035%	0.00000033%	0.00000035%
Ga-68 half life	T_1/2_ Ga-68: 62–74 min	66.1 min	64 min	62.5 min
Stability	RCP% > 90% within 240 min	RCP% > 90%	RCP% > 90%	RCP% > 90%
Ethanol content	<10% (*v*/*v*) (<2.5 g)	5.23%	6.55%	5.68%
HEPES content	Less than 200 µg/V of HEPES in test solution	Conformed	Conformed	Conformed
Endotoxins	<17.5 IU/mL	4.85 EU/mL	4.80 EU/mL	4.95 EU/mL

## Data Availability

All data generated and analysed during this study are included in this published article.

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
