# Peer review of "Development and Validation of an Analytical HPLC Method to Assess Chemical and Radiochemical Purity of [68Ga]Ga-NODAGA-Exendin-4 Produced by a Fully Automated Method"

_molecules, 2022, doi:10.3390/molecules27020543_

Round 1

Reviewer 1 Report

Migliari et al, find the validation of analytical HPLC method to improve the [68Ga] Ga-NODAGA-exendin-4 radiosynthesis by automated method., is an interesting result in radiopharamceuticals.

My few comments are

Page 3 line in 99-100, Authors need to write more brief difference between the previous results (Migliari et al., current radiopharmaceuticals, 2021, 14, 1-0) and this research findings.

Page 6 line143, figure 3., UV-HPLC condition require to add in figure so readers can easily understand.

Comments on, what are the side products (RT 5.647 and 11.517) formed?

Need to add all three validation UV-HPLC chromatograms in main article to see n compare the trice HPLC patterns.

Page 7, line 160-162, UV-HPLC appeared all peaks did not commented, I can see the more than Gallium-68 (RT 1.450) and Ga-68 bound NODAGA-exendin-4 (RT 6.955) such as RT 5.647, 6.600-700(merge to 6.947 peak) and 11.517. author need to include comments on this additional peaks.

Page 10, line 215, symbol of µ in 10 µg need to correct.

page 15, line 459, List of abbreviations not required in main article because repeated and already mentioned in main text eg. Page 1 line 29, quality controls (QCs), page 3, line 102, high-pressure liquid chromatography (HPLC) and so on.. If possible include in supporting data.

Reviewer 2 Report

The publication is certainly interesting and should be published in Molecules. Nevertheless, I have a few comments that need to be taken into account before publication:

retention time should be tR instead of Tr

Figure 1: what was the purity of solvents applied during the study?

Figure 1: please change the gradient elution profile: 0-7min – 5-100% B

Figure 2: provide standard deviations for each point at the curve

Lines 264-268: Provide exact procedure for SPE purification, what was used for conditioning, washing, sample load (the sorption capacity), elution. Please provide the data for optimization of this method or the literature source.

Lines 141-142 and Figure 3: what are “internal peaks”, why [68Ga]Ga-NODAGA-exendin-4 peak is splited? In my opinion the concentration injected to the system is to high.

Lines 163-164 and Figure 4: I don’t agree that TLC allowed for conclusion that only [68Ga]Ga-NODAGA-exendin-4 is present in the sample. The spot is so big and blurred that this result is uncertain. What is Authors opinion?

Figure 6. Instead of putting the chromatograms in frames, please make them bigger, without frames. So that they are clear to the reader (they are not now).

The Discussion is the shortest and weakest part of the publication, in places it is just a repetition of information from Results. Please change it.

Round 2

Reviewer 2 Report

The manuscript has been changed and may be accepted for publication.

Author Response

Thank you so much.

I hope my manuscript will be published.

Let me know if I have other corrections to do.

Best regards,

Silvia